# Factors associated with pulmonary tuberculosis in older adults: A scoping review

Francisco de Assis Moura Batista[1]*, Juliana Iscarlaty Freire de Araújo[1], Fernanda Cunha Soares[2‡], Thalyta Cristina Mansano Schlosser[3‡], Clarissa Terenzi Seixas[4‡], Silvana Loana Oliveira-Sousa[5‡], Gilson de Vasconcelos Torres[1‡], Ana Elza Oliveira de Mendonça[1‡], Thaiza Teixeira Xavier Nobre[6‡]

1 Department of Health Sciences, Federal University of Rio Grande do Norte, Natal, Rio Grande do Norte, Brazil, 2 Department of Dental Medicine, Division of Orthodontics and Pediatric Dentistry, Karolinska Institutet, Stockholm, Sweden, 3 Department of Nursing. State University of Campinas, Campinas, São Paulo, Brazil, 4 University Department of Nursing. Paris City University, Paris, França, 5 University of Murcia, Murcia, Espain, 6 Department of Health Sciences, Trairi Faculty of Health Sciences/Federal University of Rio Grande do Norte, Natal, Rio grande do Norte, Brazil

☯ These authors contributed equally to this work.
‡ FCS, TCMS, CTS, SLO-S, GVT, AEOM and TTXN are also contributed equally to this work.
* assisbaptista13@gmail.com

## Abstract

Tuberculosis is an infectious disease with a substantial impact on public health, with the pulmonary form representing the most significant concern. The disease poses considerable risks to the elderly population, as it frequently manifests in conjunction with other age-related conditions, thereby complicating both diagnosis and clinical management. Several factors, including comorbidities, male sex, smoking, and alcohol consumption, may contribute to the development of the disease. This study aimed to identify, through a literature review, the main factors associated with pulmonary tuberculosis in older adults. A scoping review was conducted following the guidelines of the Joanna Briggs Institute and in accordance with the PRISMA-ScR extension for this type of study and was registered with the Open Science Framework. The research question was developed using the Population, Concept, and Context framework. Searches were conducted in electronic databases including PubMed, LILACS, Web of Science, Scopus, and Embase. Additionally, gray literature was retrieved from Google Scholar and the CAPES Theses and Dissertations Catalog. Health Sciences Descriptors (DeCS), along with their MeSH and Emtree equivalents, were used to identify the terms "Older Adults," "Pulmonary Tuberculosis," and "Risk Factors," combined with Boolean operators AND and OR. Pulmonary tuberculosis was found to be strongly associated with older male individuals, those with a prior diagnosis of diabetes mellitus, smokers, alcohol users, individuals with close contact with active TB cases, and those with liver cirrhosis. The findings suggest that pulmonary tuberculosis is determined by multifactorial conditions, including those naturally associated with aging. The study underscores the need for targeted public health policies that integrate active surveillance, early diagnosis, and social support. The

**Data availability statement:** All relevant data are within the paper and its Supporting information files.

**Funding:** The author(s) received no specific funding for this work.

**Competing interests:** The authors have declared that no competing interests exist.

results provide a basis for evidence-based interventions aimed at controlling pulmonar tuberculosis in this vulnerable population.

## Introduction

Tuberculosis (TB) remains a significant public health concern both in Brazil and globally. Despite the gradual year-over-year decline in incidence and mortality rates, TB continues to rank among the ten leading causes of death worldwide, and the number of individuals who develop the disease remains high. Globally, in 2024, it was estimated that approximately 10.7 million people developed tuberculosis (TB), according to the 2025 Global Tuberculosis Report of the World Health Organization (WHO) [1].

The pulmonary form, Pulmonary Tuberculosis (PTB), and the laryngeal form are primarily responsible for sustaining the transmission chain. Patients with positive sputum smear microscopy results (bacilliferous patients) are considered highly infectious [2]. It is estimated that one person shedding bacilli over the course of a year in a community can infect, on average, 10–15 individuals. Among older adults, although TB is a preventable cause of death, the mortality rate is six times higher than in other age groups. This is likely associated with immunosuppression due to aging, malnutrition, comorbidities, and delayed diagnosis [3].

In Brazil, between 2020 and 2023, approximately 50,000 new TB cases were reported among the elderly population. Of these, 84% corresponded to the pulmonary form, while 3% presented both pulmonary and extrapulmonary forms combined. The average age of infected individuals was 69 years, with a higher incidence among males. The incidence rate among the elderly is comparable to that observed in younger adults, with both groups presenting approximately 30–40 cases per 100,000 inhabitants [4].

TB in this age group is increasingly recognized as a complex public health issue, due to specific disease characteristics and the presence of additional factors that may aggravate the clinical condition depending on the patient's overall health status. Diagnostic delays remain a persistent challenge, contributing to increased disease severity and enhancing community transmission [5].

Older adults present physiological characteristics, comorbidities, and social vulnerabilities that may influence disease detection, progression, and delayed initiation of treatment, potentially leading to more severe outcomes. Identifying the factors contributing to TB incidence and complications among the elderly is crucial for the development of more effective public health policies aimed at prevention, early detection, and appropriate management of PTB in this population.

In light of this context, a scoping review was conducted to identify, in the scientific literature, the main factors associated with the occurrence of pulmonary tuberculosis among older adults.

## Materials and methods

This scoping review was conducted in accordance with the Joanna Briggs Institute (JBI) Manual for Evidence Synthesis, which provides guidance specific to this type of study. Additionally, it followed the Preferred Reporting Items for Systematic Reviews

and Meta-Analyses extension for Scoping Reviews (PRISMA-ScR) (Uploaded as supplementary material S1 File) and was registered with the Open Science Framework (OSF) under the following DOI: 10.17605/OSF.IO/DHQVP.

A scoping review, also known as a scoping study, is a type of investigation that aims not only to explore the core concepts of a given subject but also to examine its extent, scope, and nature. Through the synthesis and dissemination of findings, this type of review seeks to identify knowledge gaps and areas where research remains limited [6].

In the first phase, the research question was developed based on the Population, Concept, and Context (PCC) framework. In this case: Population – older adults aged 60 years and above; Concept – factors associated with the development of pulmonary tuberculosis (PTB); Context – PTB diagnosis. Based on this formulation, a preliminary search of the literature was conducted to identify free-text terms and Health Sciences Descriptors (DeCS), as well as their equivalents in Medical Subject Headings (MeSH) and Emtree (Embase), using the terms "Older Adults," "Pulmonary Tuberculosis," "Risk Factors," and related terms. Boolean operators AND and OR were applied to enhance the precision of the search.

The search strategy was initially created in PubMed and later adapted to the syntax requirements of each of the databases included in the review. This strategy was designed and monitored by a research librarian. The electronic databases searched included Medline using the PubMed portal, LILACS through the Virtual Health Library (VHL), Web of Science, Scopus, and Embase through the CAPES Journals Portal. In addition, gray literature was consulted via Google Scholar and the CAPES Theses and Dissertations Catalog to ensure comprehensive coverage of relevant information. The final search of the databases was conducted on January 27, 2025, in order to ensure that all materials fell within the study's established timeframe. The full search strategies for each database and for gray literature are detailed in Table 1.

For study selection, the following inclusion criteria were adopted: (a) documents published between 2014 and 2024; (b) written in English, Portuguese, or Spanish; (c) observational, cross-sectional, or longitudinal studies; and (d) studies identifying risk factors for PTB. Exclusion criteria included: (a) studies that involved older adults but addressed only extrapulmonary TB; (b) incomplete or inconclusive studies; and (c) studies that did not directly respond to the research question.

A 10-year time frame was selected for the literature search, considering that the epidemiology of TB among older adults, disease control policies, and diagnostic methods have undergone significant updates. Very old studies may reflect healthcare contexts, operational definitions, and diagnostic technologies that differ from those currently in use. Thus, the chosen time frame aimed to ensure the inclusion of evidence aligned with contemporary epidemiology.

After conducting searches across all selected databases, study records were exported and imported into the EndNote reference management software (http://www.endnote.com/), where duplicate records were identified and removed. Once duplicates had been eliminated, the remaining studies were uploaded to Rayyan (https://www.rayyan.ai/) to blind the reviewers using the platform's blind review feature during the study selection process.

Prior to initiating data extraction, both reviewers underwent calibration to ensure consistency in study selection and data collection. This calibration consisted of jointly reading and analyzing a sample of studies, with discussions about eligibility criteria and the classification of extracted data. This phase aimed to minimize discrepancies and ensure methodological rigor and reliability.

Following this phase, the two reviewers independently screened study titles and abstracts. Discrepancies were resolved through discussion. Full-text screening of the selected documents commenced only after inter-rater agreement exceeded 75%, according to Fleiss' Kappa statistics [7]. As a result, it was not necessary to consult a third reviewer.

The data from the selected studies were analyzed in line with the review objectives. They were organized into tables, and the study selection process was documented using the PRISMA flowchart (Fig 1) for scoping reviews. Key findings were synthesized and discussed in dialogue with other relevant studies in the literature.

## Results

The search resulted in 1 669 documents, which, after the initial screening based on the primary inclusion criteria and duplicate removal, totaled 483 documents. Following the title and abstract screening stage, 12 studies were selected for

**Table 1. Search strategy conducted in each database and within the gray literature. Natal, RN, 2025.**

| Database | Search key/Search string |
|---|---|
| PUBMED | ("Tuberculosis, Pulmonary"[mh] OR "Tuberculosis, Pulmonary"[tiab] OR "Consumption*, Pulmonary"[tiab] OR "Pulmonary Consumption*"[tiab] OR "Pulmonary Phthisis"[tiab] OR "Phthises, Pulmonary"[tiab] OR "Phthisis, Pulmonary"[tiab] OR "lung tuberculosis"[tiab] OR "chronic pulmonary tuberculosis"[tiab] OR "chronic tuberculosis, lung"[tiab] OR "lung TB"[tiab] OR "pulmonary TB"[tiab] OR "tuberculous bronchitis"[tiab] OR "lung tuberculosis"[tiab]) AND ("elderly person"[tiab] OR Elder*[tiab] OR "old age"[tiab] OR "older people"[tiab] OR "aged people"[tiab] OR "oldest people"[tiab] OR "older adult*"[tiab] OR senium[tiab] OR "aged people"[tiab] OR "old adult*"[tiab] OR "oldest adult*"[tiab] OR "older person*"[tiab] OR "old person*"[tiab] OR "older patient*"[tiab] OR "oldest patient*"[tiab] OR "geriatric patient"[tiab]) |
| Scopus | TITLE-ABS-KEY=("Tuberculosis, Pulmonary" OR "Tuberculosis, Pulmonary" OR "Consumption*, Pulmonary" OR "Pulmonary Consumption*" OR "Pulmonary Phthisis" OR "Phthises, Pulmonary" OR "Phthisis, Pulmonary" OR "lung tuberculosis" OR "chronic pulmonary tuberculosis" OR "chronic tuberculosis, lung" OR "lung TB" OR "pulmonary TB" OR "tuberculous bronchitis" OR "lung tuberculosis") AND TITLE-ABS-KEY= ("elderly person" OR Elder* OR "old age" OR "older people" OR "aged people" OR "oldest people" OR "older adult*" OR senium OR "aged people" OR "old adult*" OR "oldest adult*" OR "older person*" OR "old person*" OR "older patient*" OR "oldest patient*" OR "geriatric patient") AND TITLE= ("Risk Factor*" OR "Population at Risk" OR "Risk Score*" OR "Risk Factor Score*" OR "Health Correlates" OR "Social Risk Factor*" OR "Factor, Social Risk" OR "Population at Risk" OR risk OR "relative risk" OR "risk hypothesis" OR "Risk Assessment" OR "risk analysis" OR "Analysis, Risk" OR "Assessment, Benefit-Risk" OR "Assessment, Health Risk" OR "Assessment, Risk" OR "Assessment, Risk-Benefit" OR "Assessments, Risk-Benefit" OR "Benefit Risk Assessment") |
| Embase | ('lung tuberculosis'/exp OR 'tuberculosis, pulmonary':ti,ab,kw OR 'consumption*, pulmonary':ti,ab,kw OR 'pulmonary consumption*':ti,ab,kw OR 'pulmonary phthisis':ti,ab,kw OR 'phthises, pulmonary':ti,ab,kw OR 'phthisis, pulmonary':ti,ab,kw OR 'lung tuberculosis':ti,ab,kw OR 'chronic pulmonary tuberculosis':ti,ab,kw OR 'chronic tuberculosis, lung':ti,ab,kw OR 'lung tb':ti,ab,kw OR 'pulmonary tb':ti,ab,kw OR 'tuberculous bronchitis':ti,ab,kw OR 'lung tuberculosis':ti,ab,kw) AND ('elderly person':ti,ab,kw OR 'elder*':ti,ab,kw OR 'old age':ti,ab,kw OR 'older people':ti,ab,kw OR 'aged people':ti,ab,kw OR 'oldest people':ti,ab,kw OR 'older adult*':ti,ab,kw OR 'senium':ti,ab,kw OR 'aged people':ti,ab,kw OR 'old adult*':ti,ab,kw OR 'oldest adult*':ti,ab,kw OR 'older person*':ti,ab,kw OR 'old person*':ti,ab,kw OR 'older patient*':ti,ab,kw OR 'oldest patient*':ti,ab,kw OR 'geriatric patient':ti,ab,kw) AND ('risk factor'/exp OR 'risk factor*':ti,ab,kw OR 'population at risk':ti,ab,kw OR 'risk score*':ti,ab,kw OR 'risk factor score*':ti,ab,kw OR 'health correlates':ti,ab,kw OR 'social risk factor*':ti,ab,kw OR 'factor, social risk':ti,ab,kw OR 'population at risk':ti,ab,kw OR 'risk'/exp OR 'risk':ti,ab,kw OR 'relative risk':ti,ab,kw OR 'risk hypothesis':ti,ab,kw OR 'risk assessment'/exp OR 'risk analysis':ti,ab,kw OR 'risk assessment':ti,ab,kw OR 'analysis, risk':ti,ab,kw OR 'assessment, benefit-risk':ti,ab,kw OR 'assessment, health risk':ti,ab,kw OR 'assessment, risk':ti,ab,kw OR 'assessment, risk-benefit':ti,ab,kw OR 'assessments, risk-benefit':ti,ab,kw OR 'benefit risk assessment':ti,ab,kw) |
| LILACS | (mj:"Tuberculose Pulmonar" OR ti:"Consumpção Pulmonar" OR ti:"Tuberculose do Pulmão" OR tw:"Tísica" OR tw:"Tísica Pulmona" OR tw:"Tuberculosis, Pulmonary" OR tw:"Tuberculosis Pulmonar" tw:"Tuberculosis, Pulmonary" OR tw:"Consumption*, Pulmonary" OR tw:"Pulmonary Consumption*" OR tw:"Pulmonary Phthisis" OR tw:"Phthises, Pulmonary" OR tw:"Phthisis, Pulmonary" OR ti:"lung tuberculosis" OR tw:"chronic pulmonary tuberculosis" OR tw:"chronic tuberculosis, lung" OR tw:"lung TB" OR tw:"pulmonary TB" OR tw:"tuberculous bronchitis" OR tw:"lung tuberculosis") AND (mj:idoso OR ti:idoso* OR ti:anciano OR ti:"pessoa idosa" OR tw:"Pessoas de Idade" OR tw:"elderly person" OR ti:Elder* OR tw:"old age" OR tw:"older people" OR tw:"aged people" OR tw:"oldest people" OR tw:"older adult*" OR tw:senium OR tw:"aged people" OR tw:"old adult*" OR tw:"oldest adult*" OR tw:"older person*" OR ti:"old person*" OR tw:"older patient*" OR tw:"oldest patient*" OR ti:"geriatric patient") |
| Web Of Sciene | ("Tuberculosis, Pulmonary" OR "Tuberculosis, Pulmonary" OR "Consumption*, Pulmonary" OR "Pulmonary Consumption*" OR "Pulmonary Phthisis" OR "Phthises, Pulmonary" OR "Phthisis, Pulmonary" OR "lung tuberculosis" OR "chronic pulmonary tuberculosis" OR "chronic tuberculosis, lung" OR "lung TB" OR "pulmonary TB" OR "tuberculous bronchitis" OR "lung tuberculosis") AND ("elderly person" OR Elder* OR "old age" OR "older people" OR "aged people" OR "oldest people" OR "older adult*" OR senium OR "aged people" OR "old adult*" OR "oldest adult*" OR "older person*" OR "old person*" OR "older patient*" OR "oldest patient*" OR "geriatric patient") (Topic) |
| Google Scholar | ("Tuberculosis, Pulmonary" OR "lung tuberculosis" OR "chronic pulmonary tuberculosis") AND (elderly OR aged OR "old age" OR "aged people" OR "older patient*" OR "old adult") |
| BDTD CAPES | Idoso AND tuberculose pulmonar |

Developed by authors, 2025.

full-text review, of which 6 were ultimately included in this scoping review. Fig 1 presents, in accordance with the PRISMA flowchart, the selection process for the publications included in this review.

The selected studies were published in both national and international scientific journals. The research was conducted in Brazil (n = 3) and China (n = 3). Coincidentally, the 2025 Global TB Report from the WHO indicated that among the eight countries responsible for more than two-thirds of TB cases in 2024, China accounted for approximately 6.5% of cases.

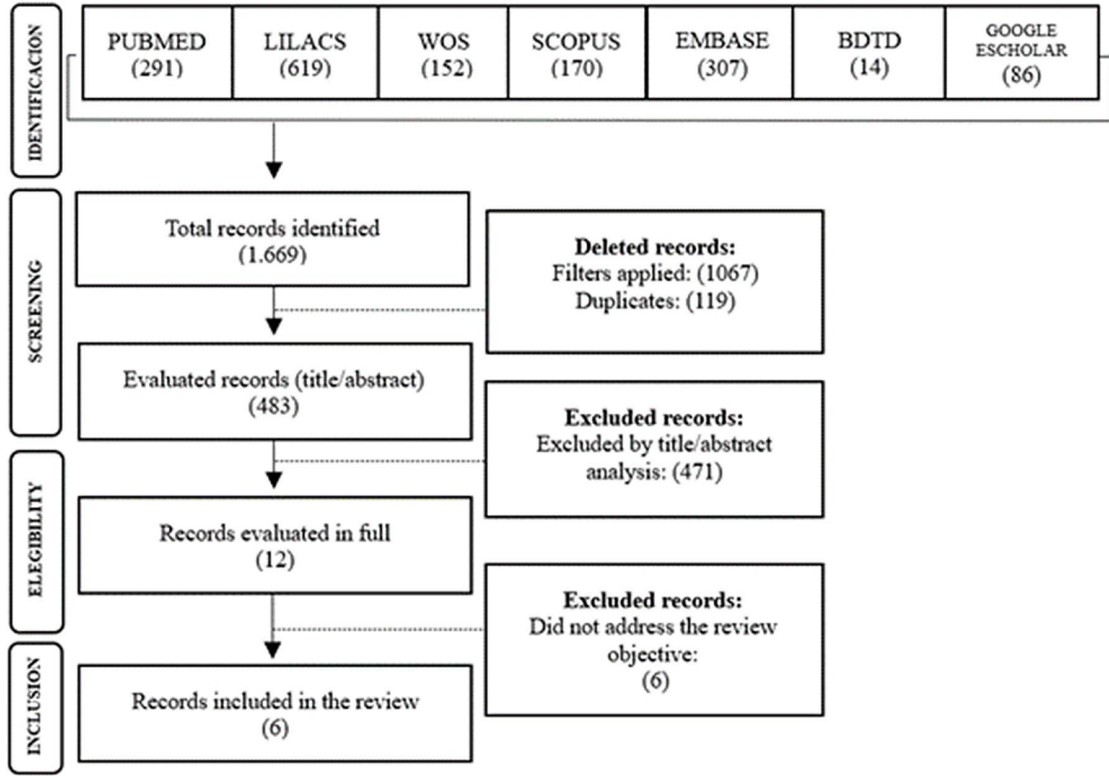

**Fig 1. Flowchart of selected studies, Natal, RN, Brazil, 2025.** Developed by authors, 2025.

In the Americas Brazil reported over 90,000 cases of the disease and estimated more than 100,000 [1]. This may directly reflect these countries' heightened concern in producing research on infection by the disease-causing agent.

Regarding the publication year, 50% of the included studies were published in 2017, with only one study published in the past five years (2024). The majority of the studies were cross-sectional in design.

The scoping review revealed a low number of studies addressing the main factors associated with the development of PTB among older adults, underscoring the limited discussion of this disease in the elderly population despite its epidemiological importance for this group.

The diagnosis of PTB in patients included in these studies followed established protocols combining clinical, epidemiological, and complementary diagnostic criteria. Patients were screened and, in addition to clinical-epidemiological criteria, underwent confirmatory diagnostic procedures such as chest X-ray [8] and laboratory tests. Three sputum samples were collected, and PTB was confirmed through bacteriologically positive sputum smear microscopy [8,9]. Data were collected secondarily from medical records or patient registries documenting TB cases during the course of the disease [10–13].

The studies confirmed that smoking among older adults [8,11,14] was strongly associated with the development of PTB, as was a prior diagnosis of diabetes mellitus (DM) [9,11,12,14], liver cirrhosis [9], significant weight loss or low body mass index (BMI) [8,9,11,12], alcohol consumption [8,11,14], chronic obstructive pulmonary disease (COPD) [12], low household income, close contact with active TB cases, and previous history of TB [9]. Most older adults in these studies were between 60 and 69 years of age [14].

All studies demonstrated that being male [8–12,14] was directly associated with the development of PTB among older adults. In one of the studies, population attributable fractions were calculated, highlighting the public health significance of these factors, with male sex presenting the highest attributable fraction [9].

Details on authorship, year of publication, journal, study type, objectives, research question, participants (including sample size), and main findings of these studies are provided in Table 2 and Table 3 for better visualization of the results.

## Discussion

The results of this scoping review highlight important aspects related to the factors associated with PTB in older adults, reinforcing established evidence in the broader literature across different age groups and underscoring the need for targeted approaches specific to this population.

Smoking is a well-known risk factor predisposing individuals to TB infection [8,11,14]. This finding is consistent with previous studies in the general population, regardless of age group [13,15]. Cigarette smoke contributes to TB pathogenesis by inducing ciliary dysfunction, reducing immune responses, and impairing macrophage activity. It may also be associated with decreased CD4 + T-cell counts, thereby increasing susceptibility to *Mycobacterium tuberculosis* infection [16].

Among the observed risk factors, male sex remained strongly associated with PTB in older adults [8–12,14]. In a prospective cohort study conducted by Jiang et al. [17] in China, which followed 246 individuals who developed TB over seven years, the incidence rate was significantly higher among men than women. Horton et al. [18] also identified male sex as a major risk factor for TB in both high- and low-income countries. Men often struggle to recognize or address their health needs, and many conditions could be prevented through timely primary care interventions [19]. Despite their increased vulnerability and higher morbidity and mortality rates, men do not engage with health services at the same rate as women [20,21].

Older adults with a prior diagnosis of DM are at increased risk of developing PTB [9,11,12,14]. In this group, latent TB infection is more likely to progress to active disease, and previous TB diagnosis further increases the risk of reactivation [22]. A systematic review by Bodke, Wagh, and Kakar [23] confirmed that DM, along with other host characteristics, significantly amplifies TB risk. Poor glycemic control may lead to various complications, including heightened susceptibility to infections. Hyperglycemia and cellular insulin deficiency impair macrophage and lymphocyte function, thereby facilitating TB progression [24].

One of the included studies found liver cirrhosis to be associated with active TB infection [8]. This finding aligns with that of Meng-Shiuan et al. [25], who observed a higher proportion of liver cirrhosis among patients with active TB, though the significance was marginal.

Significant weight loss or low BMI were also identified as predisposing factors for *M. tuberculosis* infection [8,9,11,12]. These results are consistent with previous research [26,27]. Weight loss is a key clinical sign of TB and often prompts individuals to seek medical assistance earlier [28].

**Table 2. Description of the articles included in the review according to authorship, location/year, and journal. Natal, RN, Brazil, 2025.**

| Study | Article title | Reference | Place/year | Periodical |
|---|---|---|---|---|
| S1 | Screening for pulmonary tuberculosis in type 2 diabetes elderly: a cross-sectional study in a community hospital. | Lin YH, Chen CP, Chen PY, et al. [8] | China, 2015 | BMC public health |
| S2 | Prevalence and risk factors of active pulmonary tuberculosis among elderly people in China: a population based cross-sectional study. | Zhang CY, Zhao F, Xia YY, et al. [9] | China, 2017 | Infectious diseases of poverty (BMC) |
| S3 | Aspectos epidemiológicos, clínicos e evolutivos da tuberculose em idosos de um hospital universitário de belém, Pará | Chaves EC, et al. [14] | Brazil, 2017 | Rev. Brasileira de Gerontologia |
| S4 | Aspects of tuberculosis in the elderly | Freire ILS, et al. [10] | Brazil, 2018 | Acta Scientiarum. Health Sciences |
| S5 | História clínica e epidemiológica de pacientes idosos diagnosticados com tuberculose | Vasconcelos JC. [11] | Brazil, 2017 | BDTB |
| S6 | A Study on Risk Factors for Readmission of Elderly Patients with Pulmonary Tuberculosis Within One Month Using Propensity Score Matching Method | Feng Y, Guo J, Luo S, Zhang Z, Liu Z. [12] | China, 2024 | Infection and Drug Resistance |

Research data, 2025.

**Table 3. Description of articles included in the review according to objective, study type, research question, population/sample, and main results. Natal, RN, Brazil, 2025.**

| Study | Objective | Study Design | Research Question | Population (Sample Size) | Key Findings |
|---|---|---|---|---|---|
| S1 | To actively screen high-risk elderly diabetic patients and identify TB prevalence and its determinants. | Single-center cross-sectional study | Not specified | A total of 3,087 patients with type 2 diabetes aged >65 years (2,141 hospital-based and 946 community-based) participated in the screening program. | Male sex, smoking, liver cirrhosis, and self-reported weight loss were significantly associated with increased TB risk. Independent risk factors included weight loss, cirrhosis, and smoking history. Among the 73 patients with active TB or TB history, they were more likely to be male, have lower BMI, greater alcohol consumption, a family history of TB, higher LDL levels, and less hypertension. No significant difference was found in HbA1c levels between groups. |
| S2 | To determine the prevalence and identify TB risk factors among older adults in order to develop a screening algorithm for this high-risk population in China. | Cross-sectional cluster sampling study | Not specified | Sample size was estimated using a formula for population proportion ($p = 369/100,000$, CI = 95%, error = 0.2), resulting in a required sample of 33,192 (allowing 10% non-response). | Of 38,888 eligible individuals from 27 clusters, 34,269 completed the questionnaire and physical exam. There were 193 active PTB cases, 62 bacteriologically confirmed. Estimated prevalence of active and confirmed PTB in those aged ≥65 was 563.19 per 100,000. Risk factors: male sex, older age, rural residence, low weight, diabetes, close contact with TB cases, and prior TB history. TB risk increased with age and decreased BMI in a dose-response pattern. |
| S3 | To evaluate the epidemiological, clinical, and outcome aspects of TB in older adults at a university hospital in Belém, Pará. | Cross-sectional study | Not specified | 82 medical records reviewed. | Most patients were male (64.6%), aged 60–69 years, with new TB cases (95.1%), primarily pulmonary form (75.6%), and comorbidities (69.5%). Hospital stays exceeded 21 days. Common symptoms: fever (67.1%), dyspnea (64.6%), weight loss (61.0%), productive cough (59.8%), chest pain (51.2%). Adverse effects occurred in 50% of patients, mostly gastrointestinal (70.7%). Cure rate: 59.8%; TB-related mortality was high (15.9%). Significant associations: age group ($p = 0.017$), hospitalization time ($p = 0.000$), and adverse reaction ($p = 0.018$). |
| S4 | To describe the clinical, diagnostic, and therapeutic aspects of TB in older adults. | Exploratory descriptive cross-sectional study with a quantitative approach | What are the clinical, diagnostic, and therapeutic aspects of TB in older adults treated in a health district in Natal/RN? | 94 participants aged 60–69 years. | Most participants were male (51.1%), aged 60–69. Pulmonary TB was predominant (86.2%), and most were new cases (59.6%). Treatment was self-administered (52.1%) and completed on time (57.4%). Tuberculin skin test was not performed in 76.6%, nor histopathology in 86.2%; however, chest radiography suggested TB in 72.3%. First sputum smear was not performed in 59.5%; of those tested with a second smear, 54.2% were positive. HIV testing was not performed in 54.2% of patients. |
| S5 | To study the clinical and epidemiological history of elderly TB patients in the state of Ceará. | Retrospective, documentary, quantitative, and analytical study | Not specified | 128 medical records reviewed. Patient ages ranged from 60 to 93 (mean = 68.84; variance = 52.21; SD = 7.22). | Pulmonary TB was the most common form (102; 79.69%). Smoking was the most prevalent risk factor (63; 49.22%), followed by alcohol use, HIV/AIDS, malnutrition, diabetes, and COPD. Causal links were established between these factors and mortality. Of the patients, 62 (48.44%) were discharged improved, 9 (7.03%) were transferred, and 57 (44.53%) died. Among deaths, 43 (47.78%) were male. Key symptoms: fever, dyspnea, productive cough, and weight loss. Most patients were male (70.31%), aged 60–69 (62.5%), and 46.09% had income above one minimum wage. Those with no education or incomplete elementary education were most affected. |

*(Continued)*

**Table 3.** (Continued)

| Study | Objective | Study Design | Research Question | Population (Sample Size) | Key Findings |
|---|---|---|---|---|---|
| S6 | To explore risk factors for readmission of elderly PTB patients within one month using propensity score matching (PSM). | Clinical study | What are the risk factors for readmission of elderly PTB patients one month after discharge? | Of 1,360 hospitalized elderly PTB patients, 36 had incomplete data, 40 were readmitted for unrelated reasons, and 16 died, leaving 1,268 for analysis. | After matching, no significant differences were found between groups in sex, age, occupation, BMI, or medical history (all p > 0.05). Multivariate logistic regression identified infection, drug-induced liver injury (DILI), acute heart failure (AHF), chronic kidney disease (CKD), and extrapulmonary TB (EPTB) as significant risk factors for readmission. |

Research data. 2025.

Heavy alcohol consumption was also found to be associated with increased PTB risk [8,11,14]. Excessive alcohol use may impact both TB incidence and disease progression. Individuals with alcohol use disorder are more susceptible to TB, partly because high alcohol intake is linked to cavitary pulmonary lesions on radiographic imaging and positive sputum smear microscopy [29,30].

This review also identified a relationship between chronic obstructive pulmonary disease (COPD) [12], lower household income, close contact with active TB cases, and a history of prior TB [14] as significant risk factors for TB infection in older adults. Factors such as place of residence, smoking, COPD, and unauthorized discharge have been associated with increased risk of unplanned hospital readmission for TB patients [31].

This scoping review presents several limitations. Despite employing a comprehensive search strategy across major databases, some relevant studies may have been missed due to language and publication date restrictions. Although the published review protocol indicated that both associated and causal factors would be explored, the analysis was limited to associated factors due to the small number of available studies. It was also observed that few studies fully met the established inclusion criteria, which indicates that the specific investigation of factors associated with PTB in older adults, in isolation, remains underexplored in recent scientific literature.

Nonetheless, this study presents notable strengths, including a broad synthesis of the available evidence, identification of key gaps in the literature, and guidance for future research. By consolidating data on clinical, sociodemographic, and environmental determinants, this scoping review contributes to the development of more effective strategies for the prevention, diagnosis, and management of TB in this vulnerable population.

## Conclusion

The findings of this scoping review reveal that PTB in older adults is associated with a multifactorial set of determinants, including comorbidities such as diabetes mellitus and low BMI, immunosenescence, unfavorable socioeconomic conditions, and sex-related disparities, with male sex being particularly influential. These results underscore the need for public health policies specifically designed for this age group, integrating active surveillance, early diagnosis, and intersectoral approaches that ensure social and nutritional support.

The knowledge generated by this review provides a foundation for evidence-based public health strategies and targeted interventions that not only address TBP among older adults but also promote healthy and equitable aging.

## Supporting information

**S1 File. PRISMA extension for scoping reviews 2018.** Checklist prism protocol, Natal, RN, Brazil, 2025.
(DOCX)

## Author contributions

**Conceptualization:** Francisco de Assis Moura Batista.

**Data curation:** Francisco de Assis Moura Batista, Juliana Iscarlaty Freire de Araújo.

**Formal analysis:** Francisco de Assis Moura Batista, Juliana Iscarlaty Freire de Araújo.

**Investigation:** Francisco de Assis Moura Batista.

**Methodology:** Francisco de Assis Moura Batista, Juliana Iscarlaty Freire de Araújo.

**Supervision:** Thaiza Teixeira Xavier Nobre.

**Writing – original draft:** Francisco de Assis Moura Batista, Juliana Iscarlaty Freire de Araújo.

**Writing – review & editing:** Francisco de Assis Moura Batista, Juliana Iscarlaty Freire de Araújo, Fernanda Cunha Soares, Thalyta Cristina Mansano Schlosser, Clarissa Terenzi Seixas, Silvana Loana Oliveira-Sousa, Gilson de Vasconcelos Torres, Ana Elza Oliveira de Mendonça, Thaiza Teixeira Xavier Nobre.

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
