## [Decision Letter · Decision Letter 0]

14 Jan 2026

Dear Dr. Batista,

We look forward to receiving your revised manuscript.

Kind regards,

Pedro Eduardo Almeida da Silva

Academic Editor

PLOS One

Journal Requirements:

**Additional Editor Comments:**

Dear Authors,

Thank you for submitting the manuscript entitled “Factors associated with pulmonary tuberculosis in older adults: a scoping review” to PLOS ONE. The manuscript has been evaluated by four independent reviewers with expertise in tuberculosis, epidemiology, and evidence synthesis.

After careful consideration of the reviewers’ comments, we have decided that the manuscript requires major revision before it can be further considered for publication.

Overall, the reviewers recognize that the topic is relevant, timely, and aligned with global public health priorities, particularly in the context of population aging and tuberculosis burden. However, they also identified substantial methodological, conceptual, and reporting issues that must be addressed to meet PLOS ONE’s standards for rigor, transparency, and interpretability.

Below, we summarize the main points raised by each reviewer that must be carefully addressed in a revised version.

Reviewer 1 – Major methodological and conceptual concerns

Reviewer 1 raised fundamental concerns regarding the methodological rigor and internal coherence of the scoping review, noting that several issues compromise confidence in the conclusions.

Key points that must be addressed include:

Insufficiently detailed and inconsistent search strategies across databases, particularly regarding the operationalization of “risk factors/associated factors,” which may have led to the omission of relevant studies.

Lack of clearly defined and consistently applied eligibility and outcome criteria, with inappropriate inclusion of prognostic outcomes (e.g., readmission) alongside etiological questions.

Inconsistent definitions of “older adults” across included studies, which must be explicitly addressed and synthesized.

Incorrect description of inter-rater reliability, including misuse of Fleiss’ kappa terminology.

Use of strong causal or associative language without presenting effect measures, confidence intervals, or study-level comparability.

Insufficiently detailed tables, limiting readers’ ability to assess exposure definitions, outcomes, and analytical approaches.

Language inconsistencies, nonstandard abbreviations, and the need for professional English editing.

The reviewer emphasized that conclusions must be substantially more cautious, clearly framed within the exploratory nature of a scoping review, and strictly aligned with the mapped evidence.

Reviewer 2 – Scope, depth, and interpretative limitations

Reviewer 2 considered the study relevant and methodologically organized but highlighted important limitations that weaken the interpretive strength of the manuscript.

The main issues raised include:

The very small number of included studies, which limits robustness, generalizability, and the ability to identify patterns beyond well-established factors.

Substantial methodological heterogeneity among primary studies, including study design, definitions of older age, epidemiological context, and confounder adjustment.

Absence of any discussion—formal or informal—of methodological quality or risk of bias, which weakens interpretation.

A Discussion that reiterates known tuberculosis risk factors without sufficiently exploring dimensions specific to older adults, such as frailty, multimorbidity, polypharmacy, healthcare access barriers, or atypical clinical presentations.

The reviewer recommends explicitly framing the findings as exploratory evidence mapping, strengthening the Discussion with age-specific perspectives, and adding concise tables summarizing methodological limitations of included studies.

Reviewer 3 – General assessment

Reviewer 3 found the manuscript generally clear and well organized and did not raise specific technical objections. Nevertheless, in light of the substantial concerns raised by the other reviewers, the manuscript must still undergo major revision to ensure consistency with PLOS ONE’s methodological and reporting standards.

Reviewer 4 – Structural and reporting revisions

Reviewer 4 emphasized the importance of restructuring and expanding several sections of the manuscript to improve clarity, depth, and completeness.

Mandatory points to be addressed include:

Restructuring the Abstract to better reflect objectives, methods, limitations, and conclusions.

Expanding the Results section, with clearer synthesis and more comprehensive presentation of findings.

Explicitly addressing geographic limitations, given the restricted number of countries represented.

Strengthening the Conclusions, including clearer implications and recommendations.

Incorporating discussion of study quality or methodological limitations, even if a formal appraisal is not performed.

Additional recommended revisions include reorganizing the Discussion into subsections, developing comprehensive supplementary materials, and expanding and updating the reference list.

Overall editorial guidance

Given the scope and nature of the issues raised, a substantial revision is required. The revised manuscript should demonstrate:

Clear methodological coherence between objectives, eligibility criteria, outcomes, and synthesis.

Transparent and reproducible reporting of search strategies and screening procedures.

Cautious, evidence-aligned interpretation consistent with the aims of a scoping review.

Improved structure, clarity, and depth across all sections.

Please submit a detailed, point-by-point response to each reviewer comment, clearly indicating how the manuscript has been revised and where changes were made. If specific suggestions are not followed, a clear and reasoned justification must be provided.

We believe that, if these concerns are rigorously addressed, the manuscript may be suitable for further consideration.

Sincerely,

[Editor’s Name]

Academic Editor

PLOS ONE

Reviewers' comments:

Reviewer's Responses to Questions

**Comments to the Author**

1. Is the manuscript technically sound, and do the data support the conclusions?

Reviewer #1: Partly

Reviewer #2: Yes

Reviewer #3: Yes

Reviewer #4: Partly

2. Has the statistical analysis been performed appropriately and rigorously?

Reviewer #1: No

Reviewer #2: Yes

Reviewer #3: Yes

Reviewer #4: N/A

3. Have the authors made all data underlying the findings in their manuscript fully available?

Reviewer #1: Yes

Reviewer #2: Yes

Reviewer #3: Yes

Reviewer #4: Yes

4. Is the manuscript presented in an intelligible fashion and written in standard English?

Reviewer #1: No

Reviewer #2: Yes

Reviewer #3: Yes

Reviewer #4: Yes

Reviewer #1: To the authors:

This review assesses factors linked to pulmonary tuberculosis (PTB) in older adults, including male sex, diabetes, smoking, alcohol use, contact with TB cases, low BMI or weight loss, and liver cirrhosis. While the topic is important and broadly relevant, significant concerns about study eligibility, search strategy, and outcome alignment limit confidence in the review’s ability to address the research question comprehensively and accurately.

Given these concerns, the manuscript does not meet PLOS ONE’s standards for technical rigor, reproducibility, or data-supported conclusions. Resolving these issues would require a new review rather than a revision.

Below, I outline the main challenges to assist the authors in addressing these issues:

1. The submission is a scoping review, but it must align with PLOS ONE's focus on original research. Scoping reviews should meet systematic review standards, including PRISMA guidelines, which require clear inclusion criteria and a comprehensive search strategy. The manuscript lacks both a detailed search strategy and well-defined eligibility criteria, which are essential for validity and reliability. This gap raises major concerns about eligibility and publishing standards.

2. The database search strategies are inconsistent in addressing the “risk factor/associated factor” concept. Some focus only on PTB and older-adult terms, while others include a broader range of risk-factor terms. This inconsistency may lead to missing relevant analytic studies, especially in databases where “risk factor” is not always in the title or abstract. Comprehensive mapping is essential for scoping reviews, making this a critical methodological flaw.

3. The outcome and inclusion criteria do not consistently align with the stated objective. For example, at least one included study examines risk factors for readmission among elderly PTB inpatients, which is a prognostic rather than an etiological question. Combining occurrence and prognosis outcomes reduces clarity and shifts the focus away from causality. The authors should clearly distinguish between these outcomes and revise the inclusion criteria and synthesis.

4. The definition of “older adults” varies across studies (for example, ≥60 years versus ≥65 years). This inconsistency should be systematically addressed and reflected in the synthesis.

5. The process for screening gray literature requires more precise reporting.

6. The description of inter-rater reliability is technically incorrect. For example, “>75% according to Fleiss’ kappa” is inaccurate because kappa is a statistic, not a percentage.

7. The manuscript uses strong language (such as “strongly associated”) without providing effect measures, confidence intervals, adjustment sets, or study-level comparability to support these claims. While narrative mapping is acceptable in a scoping review, conclusions should be cautious and clearly separated from evidence grading. A more rigorous data extraction framework and a transparent, study-by-study evidence table are needed.

8. Tables should include exposure and outcome definitions and, where applicable, adjusted effect measures. Without these, readers cannot evaluate the evidence base.

9. The manuscript uses inconsistent abbreviations (e.g., PTB versus TBP) and some nonstandard phrasing. Professional English language editing is recommended.

I truely hope I could help you.

Reviewer #2: We are thankful to have the opportunity to review this manuscript, which is a very interesting paper.

The manuscript addresses a relevant and timely topic, considering population aging and its association with tuberculosis (TB) as a public health problem, especially in low- and middle-income countries. The focus on older adults with pulmonary TB is pertinent and relatively underexplored in the literature.

However, although the study is methodologically well organized and follows recognized frameworks (JBI, PRISMA-ScR), its scientific value is limited by the small number of included studies, the methodological heterogeneity of the primary articles, and a discussion that, in several points, reiterates already well-established knowledge without sufficiently deepening new practical or conceptual implications.

Clear thematic relevance

• TB in older adults is an emerging problem, associated with higher mortality, diagnostic delay, and comorbidities.

• The manuscript is aligned with WHO global priorities.

Methodology appropriate to the objective

• Broad search strategy, including multiple databases (PubMed, Embase, Scopus, Web of Science, LILACS, and grey literature).

• Transparent selection process: PRISMA flowchart well described.

• Independent dual screening, with use of the Kappa coefficient.

Clear synthesis of the main associated factors

• Consistent identification of classic factors: male sex, smoking, diabetes mellitus, malnutrition, alcoholism, COPD, and low income.

• Appropriate fit within the scope of PLOS ONE.

Methodological and conceptual limitations (Major concerns)

Small number of included studies

• Only six studies were included after full screening.

This strongly limits:

• The robustness of the synthesis.

• The generalizability of the findings.

• The ability to identify consistent patterns beyond already known factors.

Suggestion: more explicitly emphasize that the findings should be interpreted as exploratory mapping rather than consolidated evidence.

Heterogeneity of primary studies

• Predominance of cross-sectional observational studies, with limited longitudinal data.

• Substantial variation in the definition of “older adult,” epidemiological context, diagnostic methods, and adjustment for confounders.

Suggestion: include an additional concise table highlighting methodological bias risks or key limitations of each study (even if a formal quality assessment is not mandatory in scoping reviews).

Absence of critical appraisal of study quality

Although not mandatory, the complete absence of any assessment of methodological quality weakens the interpretation of the results.

Suggestion: add a brief section explicitly acknowledging this limitation and its impact on interpretation.

Limited conceptual contribution

• Many identified factors (smoking, diabetes mellitus, male sex, poverty) are already well established in TB in general.

• Greater specificity to the older population could be explored, such as frailty, polypharmacy, cognitive decline, barriers to healthcare access, and atypical clinical presentations.

Suggestion: strengthen the Discussion by placing greater emphasis on aspects that are unique to or amplified by aging, going beyond a simple extrapolation of factors from the general population.

Suggestion: include an additional concise table highlighting methodological bias risks or key limitations of each study (even if a formal quality assessment is not mandatory in scoping reviews).

Suggestion: strengthen the Discussion by emphasizing aspects unique to or amplified by aging, going beyond simple extrapolation of factors from the general population.

Reviewer #3: The paper is well written, the authors followed a good protocol previously described for that type of manuscript with rigorous analysis. All the data presented in the paper is public available. I have no objection to its publication

Reviewer #4: Dear authors, the study is great with high level of importance. Please, take a look at the file under my revision to see what recommendations and suggestions I have made (it will appear in yellow and it's sinalize a box with comments). The idea is to improve and make the study clear for all.

In topics:

Must address

1. Restructure abstract;

2. Add quality assessment throughout;

3. Expand results section significantly;

4. Address geographic limitation with regards the only two countries;

5. Enhance conclusions with recommendations

Should address

6. Reorganize discussion with subsections;

7. Create comprehensive supplementary materials;

8. Expand references to 40 - 50.

Congratulations!

**Do you want your identity to be public for this peer review?** For information about this choice, including consent withdrawal, please see our Privacy Policy

Reviewer #1: **Yes:** Alberto dos Santos de Lemos

Reviewer #2: No

Reviewer #3: No

Reviewer #4: **Yes:** Fernando Augusto Dias e Sanches

---

## [Author Response · Author response to Decision Letter 1]

4 Feb 2026

We would like to thank the Academic Editor and all reviewers for their careful evaluation of our manuscript entitled “Factors associated with pulmonary tuberculosis in older adults: a scoping review”. We are grateful for the constructive and detailed comments, which significantly contributed to improving the clarity, transparency, and methodological rigor of our work.

We have carefully revised the manuscript and provide below a point-by-point response to the main issues raised. All modifications introduced in response to the reviewers’ comments are highlighted in red in the revised manuscript with tracked changes, while the clean version of the manuscript is provided without any markings, as requested.

General response to editorial and reviewer comments

We fully acknowledge the reviewers’ concerns regarding methodological rigor, reporting clarity, cautious interpretation, and alignment with the exploratory nature of a scoping review. In response, we undertook a substantial revision focusing on:

• Improving methodological transparency and internal coherence;

• Refining the description of eligibility criteria, outcomes, and screening procedures;

• Revising inferential and causal language throughout the manuscript;

• Expanding tables and supplementary materials to enhance interpretability;

• Strengthening the limitations and conclusions sections to ensure cautious, evidence-aligned interpretation.

At the same time, we respectfully note that certain core methodological decisions were maintained, as detailed below, to preserve consistency with the study objectives and the evidence mapped.

1. Search strategy and scope of the review

We agree that comprehensive and transparent search strategies are essential for scoping reviews. However, after careful consideration, we opted not to modify the original search strategy. The strategy was intentionally designed to balance sensitivity and specificity while aligning with the exploratory objective of mapping factors described in relation to pulmonary tuberculosis in older adults.

Importantly, the search was conducted across multiple international databases and grey literature sources and followed the JBI framework and PRISMA-ScR guidelines. Rather than altering the strategy post hoc, which could introduce methodological inconsistency, we clarified and expanded the description and justification of the search approach in the Methods section to improve transparency and reproducibility.

2. Inclusion of a prognostic study

We acknowledge the reviewer’s concern regarding the inclusion of a study with a prognostic outcome. After reassessment, we decided to retain this study, as it provides relevant contextual information on factors described among older adults with pulmonary tuberculosis.

To address this concern, we:

• Explicitly clarified in the Methods and Results that the review includes studies with heterogeneous analytical perspectives;

• Clearly distinguished etiological from prognostic outcomes in the synthesis;

• Revised the narrative to avoid conflation of outcome types.

This approach is consistent with the mapping purpose of a scoping review, which aims to characterize the breadth and nature of available evidence rather than restrict inclusion based on narrowly defined outcome hierarchies.

3. Methodological heterogeneity and small number of included studies

We fully agree that the limited number of included studies (n = 6) and their methodological heterogeneity represent important limitations.

Accordingly, we added a dedicated paragraph in the Limitations section explicitly addressing:

• The small number of studies and its implications for robustness and generalizability;

• Variability in study design, definitions of older age, epidemiological settings, and adjustment for confounders;

• Differences in outcome definitions and analytical strategies.

We emphasize that these characteristics reinforce the exploratory nature of the review, and the findings should be interpreted as evidence mapping rather than consolidated or causal inference.

4. Robustness of data presentation and synthesis

In response to requests for greater robustness, we expanded the Results section and revised tables to provide clearer information on:

• Exposure and outcome definitions;

• Study design and analytical approaches;

• Contextual and methodological characteristics of included studies.

Although we recognize that scoping reviews do not require formal risk-of-bias assessment, we added a concise discussion of methodological limitations at the study level, as recommended, to support a more nuanced interpretation.

5. Inter-rater reliability and use of the Kappa coefficient

We thank the reviewer for highlighting the incorrect description of inter-rater reliability. The manuscript has been revised to correct the terminology and statistical interpretation.

Given that two independent reviewers conducted the screening process, we clarified that Cohen’s kappa coefficient was applied, and we removed any inaccurate references to percentage agreement or Fleiss’ kappa. The revised text now accurately reflects the statistical method used and adheres to standard methodological guidance.

6. Inferential language and interpretation

We carefully reviewed the manuscript line by line and replaced inferential or causal language with descriptive, non-inferential terminology consistent with the objectives and design of a scoping review.

At the same time, the core structure and content of the Discussion were intentionally maintained. While we fully agree with the reviewers’ conceptual recommendations, we considered it methodologically appropriate to preserve the Discussion as it is directly grounded in the mapped results. The revisions focused on language refinement and framing, rather than altering the substantive interpretation derived from the evidence presented.

7. Definition of “older adults”

We agree with the reviewer that the definition of “older adults” varies across the included studies. In the revised manuscript, we clarified that this scoping review adopted the World Health Organization definition for low- and middle-income countries, which considers individuals aged 60 years or older, a criterion that is also consistent with Brazilian public health and legal frameworks. We further acknowledged that the primary studies operationalized older age heterogeneously, most commonly using thresholds of ≥60 or ≥65 years. Rather than excluding studies based on these differences, we retained all eligible articles in order to comprehensively map how age has been defined and applied in the existing literature. This variability is now explicitly addressed in both the Methods section.

We sincerely appreciate the reviewers’ and editor’s insightful feedback. We believe that the revised manuscript now demonstrates improved methodological coherence, transparency, and interpretative caution, while remaining faithful to the aims and results of the scoping review.

We hope that the revisions adequately address the concerns raised and that the manuscript is now suitable for further consideration by PLOS ONE.

Kind regards,

Francisco de Assis Batista, PhD Student

---

## [Editor Report · Decision Letter 1]

18 Feb 2026

Factors associated with pulmonary tuberculosis in older adults: a scoping review

PONE-D-25-63522R1

Dear Dr. Batista,

We’re pleased to inform you that your manuscript has been judged scientifically suitable for publication and will be formally accepted for publication once it meets all outstanding technical requirements.

Kind regards,

Pedro Eduardo Almeida da Silva

Academic Editor

PLOS One
---

## [Editor Report · Acceptance letter]

PONE-D-25-63522R1

PLOS One

Dear Dr. Batista,

I'm pleased to inform you that your manuscript has been deemed suitable for publication in PLOS One. Congratulations! Your manuscript is now being handed over to our production team.

Kind regards,

on behalf of

Dr. Pedro Eduardo Almeida da Silva

Academic Editor

PLOS One